# Discrete Optimization of a Gear Pump after Tooth Root Undercutting by Means of Multi-Dimensional Logic Functions

**Marian A. Partyka** [1] and **Maria Natorska** [2,*]

[1] Institute of Electrical Power Engineering and Renewable Energy, Opole University of Technology, 76 Prószkowska St., 45-758 Opole, Poland; m.partyka@po.edu.pl

[2] Faculty of Production Engineering and Logistics, Opole University of Technology, 76 Prószkowska St., 45-758 Opole, Poland

\* Correspondence: m.natorska@po.edu.pl; Tel.: +48-77-449-80-79

**Abstract:** In this paper, the optimization of a gear pump after tooth root undercutting has been investigated; this requires the volumetric, mechanical and total efficiencies of the pump to be calculated. Due to conflict in the existing model, the total efficiency is often calculated with the assumption that the other efficiencies have acceptable values. Multiple-dimensional logical functions are an additional independent method that can be used for the optimization of a pump.

**Keywords:** multiple-dimensional logical functions; multiple-dimensional logical trees; discrete optimization; gear pomp; hydraulic-mechanical efficiency $\eta_{hm}$; volumetric efficiency $\eta_v$; total efficiency $\eta_c$; Quine–McCluskey algorithm; minimization of decision function; structural classification informations

## 1. Introduction

Hydraulic systems have the ability to transfer large amounts of power with relatively high efficiency; as a result, they are becoming more and more frequently used. Liquid flow energy generators are one of the main elements of any hydraulic system. External gear pumps are the most commonly used type in industry; they are estimated to be used in approximately 50% of installations. The common use of these units is a result of their compact and simple design, as well as their small dimensions compared to other pumping units, a high efficiency coefficient, high resistance to working medium contamination, good operational reliability and a low production cost. In addition to this, the gear units are able to operate at high speeds and, in this respect, are superior to other types of reciprocating and rotary pumps. The above mentioned advantages, as well as a total efficiency of up to 90% and high operating pressures of up to 30 MPa, have an influence on their extensive applications in the control or lubrication drive systems of both plant and machines. It is important that appropriate software is used for the design of the parametric ranked graphs and trees and to store the algorithms that are used in order to avoid exponential computational complexity in complex design situations [1]. The proper operation of the system depends on the characteristics and dynamic properties of the system or component as well as changes in the structural and/or operational parameters [2–4].

## 2. Research on a Hydraulic Gear Pump with Undercut Tooth Roots

In order to optimize a gear pump, its efficiencies must be calculated: hydraulic-mechanical ($\eta_{hm}$), volumetric ($\eta_v$) and total ($\eta_c$). Taking into account the conflict in the existing model, the total efficiency is often calculated using the direct product, assuming that the remaining efficiencies are met; this leads to a large possibility for saving energy.

The total efficiency of a pump is defined by the ratio of the power output $\left(N_{wy}\right)$ to the power input $(N_{we})$ which can be written as [4–8]:

$$\eta_c = \frac{N_{wy}}{N_{we}} \cong \eta_v \times \eta_{hm} \tag{1}$$

The volumetric efficiency of a gear pump $(\eta_v)$ is determined by the ratio of the actual capacity $Q_{rz}$ to the theoretical capacity $Q_t$:

$$\eta_v = \frac{Q_{rz}}{Q_t} \tag{2}$$

The total volumetric losses in the pump are determined by the following parameters: compressibility of the liquid, deformation of the pump's components, internal leaks, the liquid's viscosity and density, and incomplete filling of the working chambers during the suction phase. Taking into account all of the coefficients and the interactions between them, the formula for the volumetric efficiency can be obtained as follows:

$$\eta_v = 1 - c_\mu \times \frac{p}{2\pi \times \mu \times n} - c_r \times \frac{1}{n} \times \sqrt{\frac{2p}{\rho}} \times \sqrt[3]{q^{-1}} \tag{3}$$

where: $c_\mu$—coefficient, which is a function of the size and number of slots in the pump, depending on the actual capacity of the pump,

$p$—working pressure,
$q$—actual performance,
$\rho$—density of the liquid,
$n$—the pump's rotational speed, $\mu$—dynamic viscosity of the liquid,
$c_r$—coefficient that depends on the type and size of the slots and the actual capacity of the pump.

The hydraulic-mechanical efficiency of a pump $(\eta_{hm})$ is defined by the theoretical moment ratio $M_t$ to the sum of the theoretical moment $M_t$ and the hydraulic-mechanical loss moment $\Delta M$ as follows:

$$\eta_{hm} = \frac{M_t}{\Delta M + M_t} \tag{4}$$

Ultimately, the following formula can be obtained:

$$\eta_{hm} = \frac{1}{1 + c_v \times 2\pi \times \frac{\mu \times n}{p} + c_\rho \times \frac{\rho \times n^2}{2p} \times \sqrt[3]{q^2} + c_p} \tag{5}$$

where: $c_p$—coefficient that depends on the type of pump,

$c_\rho$—coefficient that mainly depends on the actual capacity of the pump,
$c_v$—coefficient that depends on the type of pump,
$p$—working pressure,
$q$—actual performance,
$\rho$—density of the liquid,
$n$—the pump's rotational speed,
$\mu$—dynamic viscosity of the liquid. Using Equation (1), (3) and (5), the equation describing the total efficiency can be obtained as follows:

$$\eta_c = \frac{1 - c_\mu \times \frac{p}{2\pi \times \mu \times n} - c_r \times \frac{1}{n} \times \sqrt{\frac{2p}{\rho}} \times \sqrt[3]{q^{-1}}}{1 + c_v \times 2\pi \times \frac{\mu \times n}{p} + c_\rho \times \frac{\rho \times n^2}{2p} \times \sqrt[3]{q^2} + c_p} \tag{6}$$

In this study, $(\eta_v)$, $(\eta_{hm})$ and $(\eta_c)$ were considered as functions, while the parameters were assumed to be decision variables: $M$, $n$, $p_t$, $Q_{rz}$. The above procedure serves the purpose of using a gear pump, after tooth root undercutting, in various systems in order to demonstrate the accuracy and correctness of the mathematical calculations, and determine the discrepancies in the calculation that result from the different algorithms that were used during the design of the gear pump:

- Determination of the maximum hydraulic-mechanical efficiency, assuming the permissible volumetric efficiency;
- Determination of the maximum volumetric efficiency, assuming the permissible hydraulic-mechanical efficiency;
- Determination of the maximum overall efficiency [4–8].
- More detailed descriptions of the analyzed parameters can be given. The rated parameter expressed in the formula has been taken into account [7,8]:

$$k = \frac{n \times \mu}{p} \tag{7}$$

Such an approach requires constant consideration of the conflicting criteria of both the hydraulic-mechanical efficiency $(\eta_{hm})$ and volumetric efficiency $(\eta_v)$.

The novelty of the prototype pump consists in the modification of the involute profile in its upper part through the so-called tooth root relief (undercut). The modification can be made by means of a cutting tool with the so-called prominence or by means of an appropriate choice of engagement correction (Figure 1) [5].

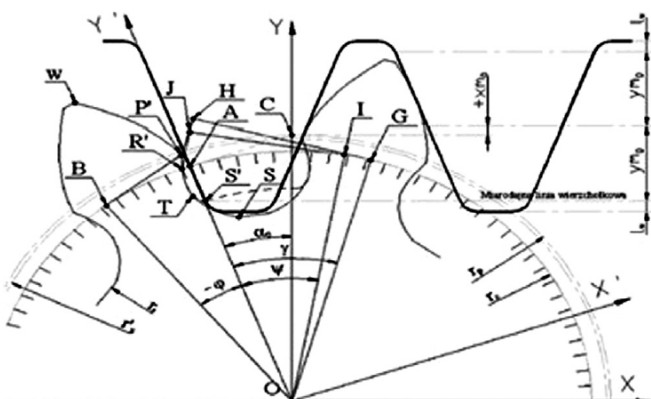

**Figure 1.** The tooth undercutting by means of a trapezoidal profile rack [5].

## 3. Discrete Optimization of a Gear Pump with Undercut Tooth Roots

For optimization of a gear pump's calculated efficiencies: hydraulic-mechanical, volumetric and total [5], the efficiency of the pump can be optimized as either mono-criteria or multi-criteria. Optimization can be carried out with variable structural and operating parameters, separately searching for the maximum efficiency value, assuming that the function of the target is the total efficiency of the pump and that the parameters that are being sought are the values of the structural and/or operating parameters [4,6]. The maximum efficiency for a pump of a particular design can be achieved by selecting the appropriate structural and operating parameters.

## 4. Logical Decision Trees

The logical structures of a decision tree contain a logical decision variable at each level of the tree which is assigned to a particular construction and/or operational parameter. Changes in the arithmetic values of the structural and/or operational parameters are coded to the branches with logical values

from left to right for individual variables and take the following values in each decision branch $p_t = 0$, 1, 2, 3, 4, 5, 6, 7, $M = 0, 1, 2, 3$ and $Q_{rz}$, $n = 0, 1, 2, 3, 4$. The layout or element design can be carried out according to any combination of parameter changes. Only decision trees with a minimum number of true branches (marked in bold), without isolated branches after graphical simplification of the full top-to-bottom nodes, give a description of the true importance of the structural and/or operational parameters, from the most important at the bottom at the root, to the least important at the top or at the tip of the tree [9]. There is a rule that that states that there can only be one decision variable per level/floor of a tree [9,10].

Arithmetic values were selected for the analysis of the examined parameters; these were then coded with the logical decision variables for the logical decision trees for the optimization of a discrete gear pump after tooth root undercutting [5]:

$n = 500$ [rpm] ~ 0; $n = 800$ [rpm] ~ 1; $n = 1000$ [rpm] ~ 2; $n = 1500$ [rpm] ~ 3; $n = 2000$ [rpm] ~ 4;
$p_t = \approx 0$ [MPa] ~ 0; $p_t = 5$ [MPa] ~ 1; $p_t = 10$ [MPa] ~ 2; $p_t = 15$ [MPa] ~ 3; $p_t = 20$ [MPa] ~ 4; $p_t = 25$ [MPa] ~ 5;
$p_t = 28$ [MPa] ~ 6; $p_t = 30$ [MPa] ~ 7;

$$Q_{rz} \in \langle 20.2; 21.1 \rangle \left[ \frac{l}{min} \right] \sim 0; \ Q_{rz} \in \langle 34.2; 34.9 \rangle \left[ \frac{l}{min} \right] \sim 1; \ Q_{rz} \in \langle 43.3; 44.5 \rangle \left[ \frac{l}{min} \right] \sim 2;$$

$$Q_{rz} \in \langle 65.5; 67.3 \rangle \left[ \frac{l}{min} \right] \sim 3; \ Q_{rz} \in \langle 87.6; 89.3 \rangle \left[ \frac{l}{min} \right] \sim 4;$$

$$M \in \langle 2.0; 47.0 \rangle \ [Nm] \ \sim 0; \ M \in \langle 77.0; 125.0 \rangle \ [Nm] \ \sim 1;$$

$$M \in \langle 138.0; 182.0 \rangle \ [Nm] \ \sim 2; M \in \langle 200.0; 259.0 \rangle \ [Nm] \ \sim 3$$

The next step was to code the logical decision variables into complex multi-value logical decision trees. Numerical values of the individual efficiency changes were adopted as follows: $\eta_v \geq 0.96$; $\eta_{hm} \geq 0.89$; $\eta_c \geq 0.86$ (Table 1) [5].

4! = 24 decision trees were drawn for each of the efficiency measures in order to obtain accurate results; this showed all of the possible combinations of decision variable swaps on four levels. The optimal arrangement was then chosen; i.e., the tree with the smallest number of true branches.

The general table of values presents all of the coded arithmetic and logical values for $M$, $n$, $p_t$, $Q_{rz}$ (Table 1), and each of the efficiency values have been selected and grouped accordingly: $\eta_v$, $\eta_{hm}$ and $\eta_c$ (Tables 2–4) [3–6].

The end result of the encoding was that the values of the variables could be applied in multi-value logical tree structures; therefore, allowing the appropriate conclusions to be obtained, as in the literature [9,10].

On the basis of the data from Table 2, the resulting logical trees (Figures 2–5) were drawn for the efficiency values of, $\eta_v$, $\eta_{hm}$, $\eta_c$ [11].

It can be proven that the best system, in terms of volumetric efficiency, hydraulic and mechanical efficiency, as well as total efficiency, in terms of a minimum number of real branches, is the floor arrangement starting from the roots $nQ_{rz}Mp_t$ and $Q_{rz}nMp_t$.

**Table 1.** Arithmetic and logical values of the established structural and/or operational parameters and target functions [5]. * Numerical values of the individual efficiency changes were adopted as follows: $\eta_v \geq 0.96$; $\eta_{hm} \geq 0.89$; $\eta_c \geq 0.86$.

| No. | $n$ | $p_t$ | $Q_{rz}$ | $M$ | $\eta_v$ [%] | $\eta_{hm}$ [%] | $\eta_c$ [%] |
|-----|-----|-------|----------|-----|--------------|-----------------|--------------|
| 1. | 0 | 0 | 0 | 0 | 94.6 | 0.0 | 0.0 |
| 2. | 0 | 1 | 0 | 0 | 92.1 | 98.0 * | 90.3 * |
| 3. | 0 | 2 | 0 | 1 | 91.3 | 91.8 * | 83.8 |
| 4. | 0 | 3 | 0 | 1 | 90.9 | 91.5 * | 83.1 |
| 5. | 0 | 4 | 0 | 2 | 90.9 | 90.7 * | 82.4 |
| 6. | 0 | 5 | 0 | 3 | 92.1 | 88.5 | 81.5 |
| 7. | 0 | 6 | 0 | 3 | 92.5 | 90.9 * | 84.1 |
| 8. | 0 | 7 | 0 | 3 | 93.0 | 90.0 * | 83.6 |
| 9. | 1 | 0 | 1 | 0 | 98.0 * | 0.0 | 0.0 |
| 10. | 1 | 1 | 1 | 0 | 97.5 * | 92.8 * | 90.5 * |
| 11. | 1 | 2 | 1 | 1 | 96.2 * | 90.6 * | 87.2 * |
| 12. | 1 | 3 | 1 | 1 | 96.0 * | 89.9 * | 86.3 * |
| 13. | 1 | 4 | 1 | 2 | 95.7 | 88.4 | 84.6 |
| 14. | 1 | 5 | 1 | 3 | 97.0 * | 87.6 | 85.0 |
| 15. | 1 | 6 | 1 | 3 | 97.5 * | 88.5 | 86.3 * |
| 16. | 1 | 7 | 1 | 3 | 97.8 * | 88.5 | 86.5 * |
| 17. | 2 | 0 | 2 | 0 | 99.9 * | 0.0 | 0.0 |
| 18. | 2 | 1 | 2 | 0 | 99.1 * | 92.8 * | 92.0 * |
| 19. | 2 | 2 | 2 | 1 | 98.7 * | 86.2 | 85.1 |
| 20. | 2 | 3 | 2 | 1 | 97.4 * | 85.6 | 83.4 |
| 21. | 2 | 4 | 2 | 2 | 97.4 * | 84.2 | 82.1 |
| 22. | 2 | 5 | 2 | 3 | 97.4 * | 85.1 | 82.9 |
| 23. | 2 | 6 | 2 | 3 | 97.4 * | 84.7 | 82.5 |
| 24. | 2 | 7 | 2 | 3 | 97.2 * | 85.3 | 82.9 |
| 25. | 3 | 0 | 3 | 0 | 100.9 * | 0.0 | 0.0 |
| 26. | 3 | 1 | 3 | 0 | 100.0 * | 84.0 | 84.0 |
| 27. | 3 | 2 | 3 | 1 | 99.6 * | 84.1 | 83.8 |
| 28. | 3 | 3 | 3 | 1 | 99.1 * | 84.9 | 84.1 |
| 29. | 3 | 4 | 3 | 2 | 98.1 * | 82.3 | 80.7 |
| 30. | 3 | 5 | 3 | 3 | 98.4 * | 84.2 | 82.9 |
| 31. | 3 | 6 | 3 | 3 | 98.2 * | 84.3 | 82.8 |
| 32. | 3 | 7 | 3 | 3 | 98.1 * | 83.3 | 81.7 |
| 33. | 4 | 0 | 4 | 0 | 100.3 * | 0.0 | 0.0 |
| 34. | 4 | 1 | 4 | 0 | 100.0 * | 75.0 | 75.0 |
| 35. | 4 | 2 | 4 | 1 | 99.3 * | 75.2 | 74.6 |
| 36. | 4 | 3 | 4 | 1 | 98.8 * | 76.9 | 76.0 |
| 37. | 4 | 4 | 4 | 2 | 98.4 * | 77.8 | 76.5 |
| 38. | 4 | 5 | 4 | 3 | 98.8 * | 82.7 | 81.7 |
| 39. | 4 | 6 | 4 | 3 | 98.7 * | 82.2 | 81.2 |
| 40. | 4 | 7 | 4 | 3 | 98.6 * | 82.0 | 80.9 |

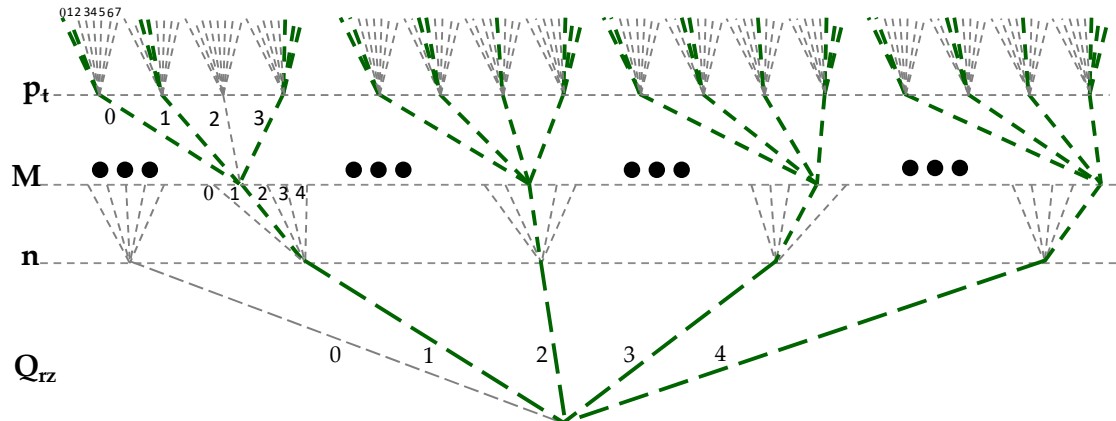

**Figure 2.** Volumetric efficiency $\eta_v$.

**Table 2.** Logically coded data for $\eta_v$. The data were from the green tree branches in Figure 2.

| | Volumetric Efficiency Data $\eta_v$ | | |
| :---: | :---: | :---: | :---: |
| **No.** | **$Q_{rz}$** | **$n$** | **$M$** | **$p_t$** |
| 9. | 1 | 1 | 0 | 0 |
| 10. | 1 | 1 | 0 | 1 |
| 11. | 1 | 1 | 1 | 2 |
| 12. | 1 | 1 | 1 | 3 |
| 14. | 1 | 1 | 3 | 5 |
| 15. | 1 | 1 | 3 | 6 |
| 16. | 1 | 1 | 3 | 7 |
| 17. | 2 | 2 | 0 | 0 |
| 18. | 2 | 2 | 0 | 1 |
| 19. | 2 | 2 | 1 | 2 |
| 20. | 2 | 2 | 1 | 3 |
| 21. | 2 | 2 | 2 | 4 |
| 22. | 2 | 2 | 3 | 5 |
| 23. | 2 | 2 | 3 | 6 |
| 24. | 2 | 2 | 3 | 7 |
| 25. | 3 | 3 | 0 | 0 |
| 26. | 3 | 3 | 0 | 1 |
| 27. | 3 | 3 | 1 | 2 |
| 28. | 3 | 3 | 1 | 3 |
| 29. | 3 | 3 | 2 | 4 |
| 30. | 3 | 3 | 3 | 5 |
| 31. | 3 | 3 | 3 | 6 |
| 32. | 3 | 3 | 3 | 7 |
| 33. | 4 | 4 | 0 | 0 |
| 34. | 4 | 4 | 1 | 1 |
| 35. | 4 | 4 | 1 | 2 |
| 36. | 4 | 4 | 2 | 3 |
| 37. | 4 | 4 | 3 | 4 |
| 38. | 4 | 4 | 3 | 5 |
| 39. | 4 | 4 | 3 | 6 |
| 40. | 4 | 4 | 3 | 7 |

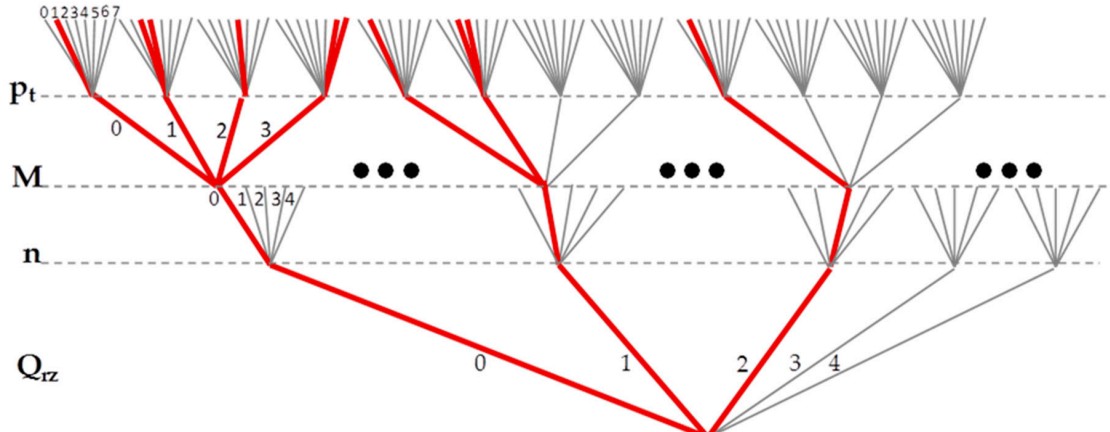

**Figure 3.** Hydraulic-mechanical efficiency $\eta_{hm}$.

**Table 3.** Logically coded data for $\eta_{hm}$. The data were from the red tree branches in Figure 3.

| Hydraulic-Mechanical Efficiency Data $\eta_{hm}$ | | | | |
|---|---|---|---|---|
| No. | $Q_{rz}$ | $n$ | $M$ | $p_t$ |
| 2. | 0 | 0 | 0 | 1 |
| 3. | 0 | 0 | 1 | 2 |
| 4. | 0 | 0 | 1 | 3 |
| 5. | 0 | 0 | 2 | 4 |
| 7. | 0 | 0 | 3 | 6 |
| 8. | 0 | 0 | 3 | 7 |
| 10. | 1 | 1 | 0 | 1 |
| 11. | 1 | 1 | 1 | 2 |
| 12. | 1 | 1 | 1 | 3 |
| 18. | 2 | 2 | 0 | 1 |

**Table 4.** Logically coded data for $\eta_c$.

| Total Efficiency Data $\eta_c$ | | | | |
|---|---|---|---|---|
| No. | $Q_{rz}$ | $n$ | $M$ | $p_t$ |
| 2. | 0 | 0 | 0 | 1 |
| 10. | 1 | 1 | 0 | 1 |
| 11. | 1 | 1 | 1 | 2 |
| 12. | 1 | 1 | 1 | 3 |
| 15. | 1 | 1 | 3 | 6 |
| 16. | 1 | 1 | 3 | 7 |
| 18. | 2 | 2 | 0 | 1 |

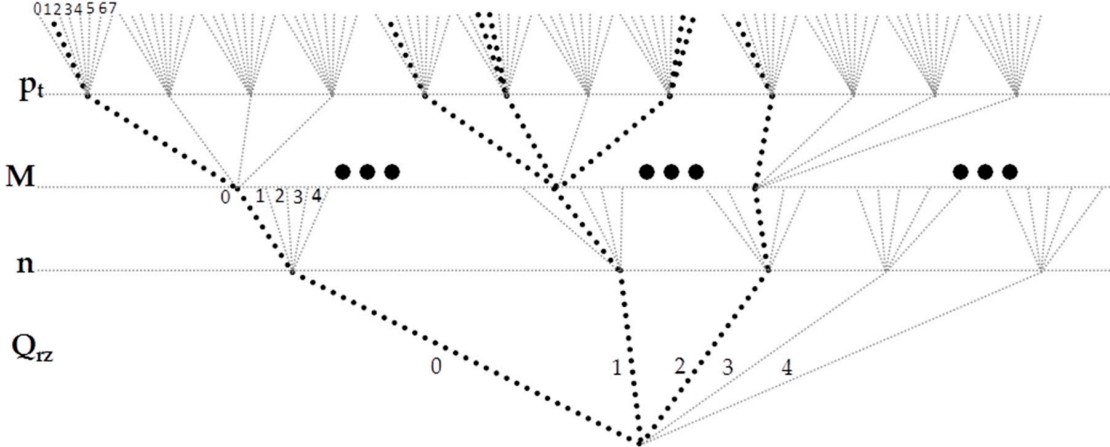

**Figure 4.** Total efficiency $\eta_c$.

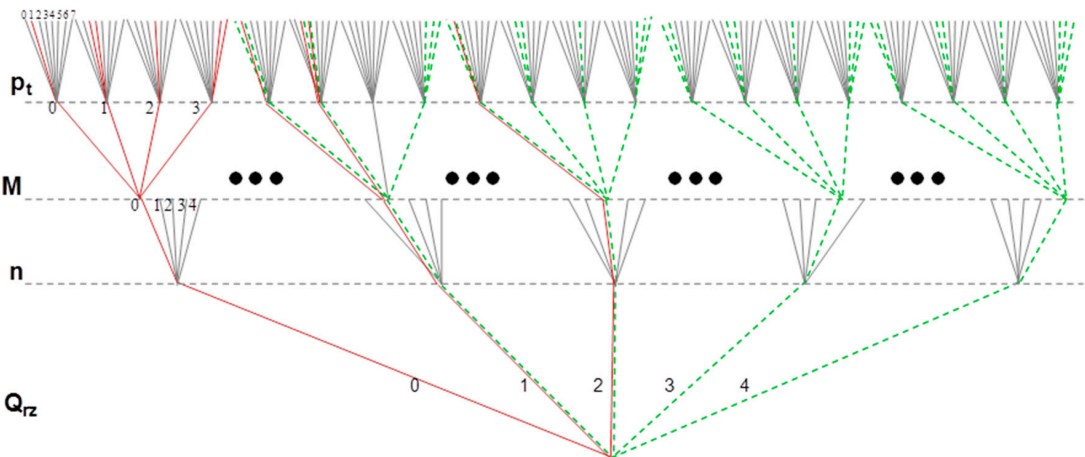

**Figure 5.** Volumetric efficiency $\eta_v$ and hydraulic-mechanical efficiency $\eta_{hm}$.

## 5. Quine–McCluskey Algorithm for Determining the Most Important Design Guidelines

The arithmetic changes in the structural and/or operational parameters, coded by branch logical values for particular variables, take the following values (Table 5):

$$n = 0, 1, 2, 3, 4.$$
$$p_t = 0, 1, 2, 3, 4, 5, 6, 7.$$
$$Q_{\text{rz}} = 0, 1, 2, 3, 4.$$
$$M = 0, 1, 2, 3.$$

Every path of the logical decision tree is a multi-value logical conjunction. The code entries are justified by the Rosser–Turquette system, provided that different multi-values of the variables exist [1,12–16].

The collection of logical functions is as follows:

- constants $0, \ldots, m - 1 = \max\{m_1 - 1, \ldots, m_n - 1\}$,
- alternative (sum)—max $(x_1, x_2)$,
- conjunction (product)—min $(x_1, x_2)$,
- characteristic function $j_i(x)$, where:

$$j_k(x_i) = \begin{cases} m - 1, x_i = k \\ 0, x_i \neq k \end{cases}$$

for $x_i = 0, \ldots, m_i - 1; i = 1, \ldots, n$ is functionally complete in $Z_{m_1, \ldots, m_n}$ (Rosser–Turquette partial system).

**Table 5.** Logical data arranged by the sum of the digits in the records of the data. (•—empty set).

| | $\eta_c$ | | | | $\eta_{hm}$ | | | | $\eta_v$ | | |
|---|---|---|---|---|---|---|---|---|---|---|---|
| $n$ | $p_t$ | $Q_{rz}$ | $M$ | $n$ | $p_t$ | $Q_{rz}$ | $M$ | $n$ | $p_t$ | $Q_{rz}$ | $M$ |
| 0 | 1 | 0 | 0 | 0 | 1 | 0 | 0 | 1 | 0 | 1 | 0 |
| | • | | | | • | | | 1 | 1 | 1 | 0 |
| 1 | 1 | 1 | 0 | 0 | 2 | 0 | 1 | 2 | 0 | 2 | 0 |
| | • | | | 1 | 1 | 1 | 0 | 1 | 2 | 1 | 1 |
| 2 | 1 | 2 | 0 | 0 | 3 | 0 | 1 | 2 | 1 | 2 | 0 |
| 1 | 2 | 1 | 1 | 1 | 2 | 1 | 1 | 1 | 3 | 1 | 1 |
| 1 | 3 | 1 | 1 | 2 | 1 | 2 | 0 | 3 | 0 | 3 | 0 |
| | • | | | 0 | 4 | 0 | 2 | 2 | 2 | 2 | 1 |
| | • | | | 1 | 3 | 1 | 1 | 3 | 1 | 3 | 0 |
| | • | | | | • | | | 2 | 3 | 2 | 1 |
| | • | | | | • | | | 4 | 0 | 4 | 0 |
| 1 | 6 | 1 | 3 | 0 | 6 | 0 | 3 | 3 | 2 | 3 | 1 |
| 1 | 7 | 1 | 3 | 0 | 7 | 0 | 3 | 4 | 1 | 4 | 0 |
| | | | | | | | | 1 | 5 | 1 | 3 |
| | | | | | | | | 2 | 4 | 2 | 2 |
| | | | | | | | | 3 | 3 | 3 | 1 |
| | | | | | | | | 1 | 6 | 1 | 3 |
| | | | | | | | | 4 | 2 | 4 | 1 |
| | | | | | | | | 1 | 7 | 1 | 3 |
| | | | | | | | | 2 | 5 | 2 | 3 |
| | | | | | | | | 3 | 4 | 3 | 2 |
| | | | | | | | | 4 | 3 | 4 | 1 |
| | | | | | | | | 2 | 6 | 2 | 3 |
| | | | | | | | | 2 | 7 | 2 | 3 |
| | | | | | | | | 3 | 5 | 3 | 3 |
| | | | | | | | | 4 | 4 | 4 | 2 |
| | | | | | | | | 3 | 6 | 3 | 3 |
| | | | | | | | | 3 | 7 | 3 | 3 |
| | | | | | | | | 4 | 5 | 4 | 3 |
| | | | | | | | | 4 | 6 | 4 | 3 |
| | | | | | | | | 4 | 7 | 4 | 3 |

In the Rosser–Turquette partial system, the following correlations occur ($x_1, \ldots, x_n$—variables):

1. $(x_1 + x_2) + x_3 = x_1 + (x_2 + x_3)$
2. $(x_1 \times x_2) \times x_3 = x_1 \times (x_2 \times x_3)$
3. $x_1 + x_2 = x_2 + x_1$
4. $x_1 \times x_2 = x_2 \times x_1$
5. $x_i + 0 = 0 + x_i = x_i; x_i \times 0 = 0 \times x_i = 0$
6. $x_i \times (m_i - 1) = (m_i - 1) \times x_i = x_i$
   $x_i + (m_i - 1) = (m_i - 1) + x_i = m_i - 1$

$$x_i \times (m-1) = (m-1) \times x_i = x_i$$
$$x_i + (m-1) = (m-1) + x_i = m-1$$

7. $\quad x_1 + (x_2 \times x_3) = (x_1 + x_2) \times (x_1 + x_3)$

8. $\quad x_1 \times (x_2 + x_3) = x_1 \times x_2 + x_1 \times x_3$

9. $\quad x_i = 0 \times j_0(x_i) + \ldots + (m_i - 1) \times j_{m_i - 1}(x_i)$

10. $\quad j_0(x_i) + \ldots + j_{m_i - 1}(x_i) = m - 1$

11. $\quad j_u(x_i) \times j_v(x_i) = \begin{cases} j_u(x_i), u = v \\ 0, u \neq v \end{cases}$

12. $\quad j_u(j_v(x_i)) = \begin{cases} j_v(x_i), u = m - 1 \\ 0, 0 < u < m - 1 \\ j_0(x_i) + \ldots + j_{v-1}(x_i) + j_{v+1}(x_i) + \ldots + j_{m_i - 1}(x_i), u = 0 \end{cases}$

13. $\quad j_u(x + y) = j_u(x) \times (j_0(y) + \ldots + j_u(y)) + j_u(y) \times (j_0(x) + \ldots + j_u(x))$

14. $\quad j_u(x \times y) = j_u(x) \times (j_u(y) + \ldots + j_{m-1}(y)) + j_u(y) \times (j_u(x) + \ldots + j_{m-1}(x))$

Some of the above correlations can be written in a different form; however, the alternative-conjunctive notation has been purposefully used, e.g.,:

$$j_0(j_v(x)) = \begin{cases} 0 \iff j_v(x) \neq 0 \iff x = v \\ m - 1 \iff j_v(x) = 0 \iff x \neq v \end{cases}$$

Setting the most important design guidelines results in a logical minimization on the multi-value logical conjunction. A procedure such as this is correct, as the logical product is an essential design guideline, assuming that each element qualitatively describes only one structural and/or operational parameter. If a product subset has identical logical values that qualitatively describe all of the structural and/or operational parameters with the exception of only one parameter, therefore, allowing all of the possible variations for the purpose of designing the system under the given operating conditions, then a logical minimization can be carried out for the parameter. From a practical perspective, such an approach produces indifference, i.e., exclusion of a given parameter, as it does not play any role in the correct operation of the system. In mathematical logic and automatics theory, the result of the minimization is written in the form of a dash (—) which indicates indifference. The Quine–McCluskey algorithm that is used to minimize the multi-valued logical functions ensures that all the dashes for a given set of design guidelines can be found from the sum of the logical products. By using dependencies, the following can be found:

$$A \times j_0(x_r) + \ldots + A \times j_{m-1}(x_r) = A$$
$$A \times j_u(x_r) + A = A$$
$$\text{where, } A = A(x_1, \ldots, x_{r-1}, x_{r+1}, \ldots, x_n),$$
$$j_u(x_r) = \begin{cases} m - 1, u = x_r \\ 0, u \neq x_r \end{cases} \quad 0 \leq u \leq m - 1$$

a dash (—) can be obtained to indicate indifference for $x_r$.

### 5.1. Example 1

The multi-value logical function $f(x_1, x_2, x_3)$ of three variables $x_1, x_2 = 0, 1, 2; x_3 = 0, 1, 2, 3, 4$, allocated to three structural and/or operational parameters $X_1, X_2, X_3$ with arithmetic values and physical dimensions, were used to code a set of design guidelines in the form of products. Where 0, 1, 2 conventionally represents "0" a decrease in the arithmetic value, "1" leaving it unchanged and "2" an increase in the arithmetic value, respectively, for $X_1, X_2$, for $X_3$ the logical values of 0, 1, 2, 3, 4 respectively represents leaving it unchanged "2", "0" and "1"—a large decrease and a decrease; "3" and "4"—an increase and a large increase in the arithmetic value.

The minimization for a partial multi-value logical function, according to the simplified version of the McCluskey algorithm, is performed in the following order:

| | | | | | | | |
|---|---|---|---|---|---|---|---|
| 020 | V | | | | | | |
| 200 | | V | | | | | |
| 101 | | | | V | | | **02–** |
| 021 | V | | | | | V | **20–** |
| 201 | | V | | | V | | **1–1** |
| 111 | | | | V | | | **21–** |
| 210 | | | V | | | | **2–1** |
| 022 | V | | | | | | **–21** |
| 202 | | V | | | | | |
| 121 | | | | V | | V | |
| 211 | | | V | | V | | |
| 023 | V | | | | | | |
| 203 | | V | | | | | |
| 212 | | | V | | | | |
| 221 | | | | | V | V | |
| 024 | V | | | | | | |
| 204 | | V | | | | | |
| 213 | | | V | | | | |
| 214 | | | V | | | | |

1. $021 + 121 + 221 = (-\,2\,1)$

   $j_0(x_1)j_2(x_2)j_1(x_3) + j_1(x_1)j_2(x_2)j_1(x_3) + j_2(x_1)j_2(x_2)j_1(x_3) = j_2(x_2)j_1(x_3);$

   •

   •

   •

6. $020 + 021 + 022 + 023 + 024 = (02-)$

   $j_0(x_1)j_2(x_2)j_0(x_3) + j_0(x_1)j_2(x_2)j_1(x_3) + j_0(x_1)j_2(x_2)j_2(x_3) + j_0(x_1)j_2(x_2)j_3(x_3) + j_0(x_1)j_2(x_2)j_4(x_3) = j_0(x_1)j_2(x_2).$

Six results were obtained from the minimization process as reduced logical products that describe the most important design guidelines, e.g.,: (1—1) leaving it unchanged for $X_1$ and freedom for $X_2$ (reduction, unchanged or increased) and a reduction for $X_3$ of the appropriate arithmetic values in order to design the system for the specified operating conditions. The formal method of logical minimization can often not be applied literally to the graphical logical decision-making processes. Due to the fact that there are isolated branches on multivalent logical trees, which are an adverse phenomenon, this means that there are interrupted decision paths from the root at the bottom, to the tips at the top.

### 5.2. Example 2

For the sum of the multi-value logical products (02–) + (20–) + (1–1) + (21–) + (2–1) + (–21), which are the most important design guidelines, a logical tree with isolated branches can be produced (Figure 6).

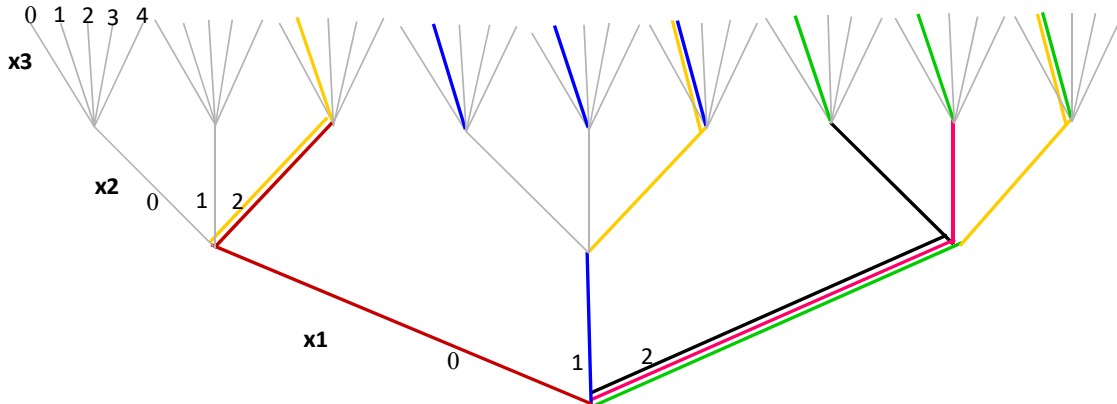

**Figure 6.** Logical tree with isolated branches.

*5.3. Example 3*

$F(x_1, x_2, x_3) = \bar{x}_1\,\bar{x}_2\,\bar{x}_3 + x_1\bar{x}_2\,\bar{x}_3 + \bar{x}_1\,\bar{x}_2\,x_3 + x_1x_2x_3 + x_1\bar{x}_2x_3 =$
$= j_0(x_1)j_0(x_2)j_0(x_3) + j_1(x_1)j_0(x_2)j_0(x_3) + j_0(x_1)j_0(x_2)j_1(x_3) + j_1(x_1)j_1(x_2)j_1(x_3) + j_1(x_1)j_0(x_2)j_1(x_3);$
$F(x_1, x_2, x_3) = \bar{x}_2 + x_1x_3 = j_0(x_2) + j_1(x_1)j_1(x_3);$
$000 + 100 + 001 + 111 + 101 = (-\,0\,-) + (1-1)$

as shown in [1] and Figure 7

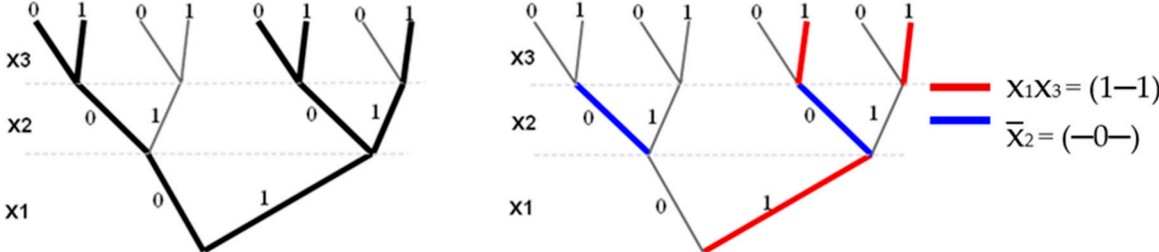

**Figure 7.** Traditional two-valued minimization.

$$\widetilde{F}(x_1, x_2, x_3) = F(x_2, x_1, x_3) = 000 + 001 + 010 + 011 + 111 = (0 - -) + 1(11)$$
$$\widetilde{F}(x_1, x_2, x_3) = F(x_2, x_1, x_3) = \overline{x_2} + x_2x_1x_3 = j_0(x_2) + j_1(x_2)(j_1(x_1)j_1(x_3))$$

as shown in [1] and Figure 8

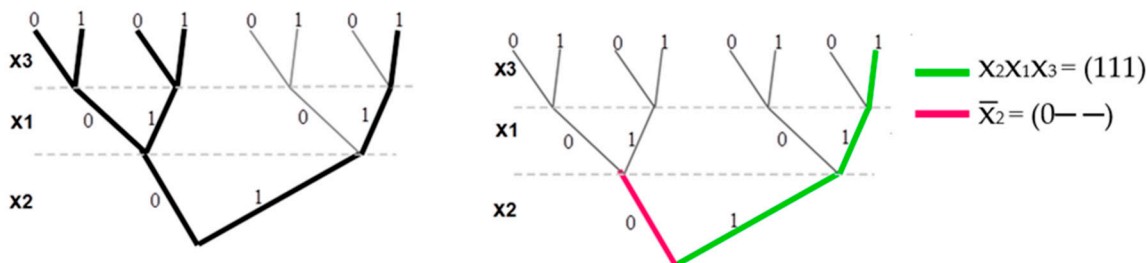

**Figure 8.** Different version of the equivalent continual two-valued decision tress.

## 6. Multi-Dimensional Decision Trees

Determining the maximum volumetric efficiency $\eta_v$, while satisfying the permissible hydraulic-mechanical efficiency $\eta_{hm}$ or determining the maximum hydraulic-mechanical efficiency $\eta_{hm}$ while satisfying the permissible volumetric efficiency $\eta_v$, concerns the same gear pump after tooth root undercutting. Therefore, instead of additionally counting the total efficiency $\eta_c$ independently,

multi-value, multi-dimensional logical trees can be created for $\eta_v$, $\eta_{hm}$ with level layouts that are identical to the best ones for $\eta_v$, $\eta_{hm}$ separately (Figure 5), which are similar to those in the literature [9].

Figure 5 shows the identical decision paths (simultaneously) for $\eta_v$, $\eta_{hm}$ and has compared them with $\eta_c$ (Figure 4) in order to produce the final optimal structural selection according to Table 1 and the corresponding level layouts $nQ_{rz}Mp_t$ and $Q_{rz}nMp_t$.

## 7. Conclusions

For the gear pump after tooth root undercutting that has been presented in this paper, there were no dashes (—) in the coded logical data; this was because the conditions for minimization were not met. Hence, no minimization was carried out and the multi-value logical trees did not have branches that were isolated for $\eta_v$, $\eta_{hm}$ Multi-dimensional logical decision trees are independent of any other complex design methods. The shared pathways that were pursued implied the fulfilment of a compromise in order to achieve an optimal solution according to a set of criteria. Any deviations in the calculation were mostly due to incorrect arithmetical computational rounding for efficiencies $\eta_v$, $\eta_{hm}$, $\eta_c$.

The best solutions for the gear pump after tooth root undercutting in this example were trees with the order of levels: $nQ_{rz}Mp_t$ and $Q_{rz}nMp_t$ and multi-dimensional decision trees (which differ by only a few branches at the top levels of the trees).

Since the trees for the layout $nQ_{rz}Mp_t$ appear the same, and as the values $Q_{rz}$ and $n$ take the same parameters (Tables 1–4), Figure 4 only shows the system $Q_{rz}nMp_t$ for efficiencies $\eta_v$, $\eta_{hm}$, $\eta_c$.

A similar analysis of multi-dimensional logical trees was performed in the literature [9]. The total efficiency $\eta_c$ was determined independently and a logical multi-dimensional tree was produced for both volumetric efficiency $\eta_v$ and hydraulic-mechanical efficiency $\eta_{hm}$ and then compared with the logical decision tree for total efficiency $\eta_c$.

More complex cases require the development of a special algorithm in order to determine the optimal multi-value multi-dimensional logic trees and multi-value logical functions; this is a task for future research.

**Author Contributions:** Writing; Review and Editing, M.A.P. and M.N. All authors have read and agreed to the published version of the manuscript.

**Funding:** This research received no external funding.

**Conflicts of Interest:** The authors declare no conflict of interest.

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
