# Peer review of "Discrete Optimization of a Gear Pump after Tooth Root Undercutting by Means of Multi-Dimensional Logic Functions"

_applsci, doi:10.3390/app10134682_

Round 1

Reviewer 1 Report

The comments are as follows:

  1. The gear design parameters and their values for the gear pumps are not clear.
  2. Please give more explanations for what the tooth root undercutting
  3. The sections 4 and 5 should been improved. The concept and algorithm are not clearly shown.
  4. How do you determine the values of n, p, Q and M in your cases?
  5. How is the effect level/ranking of efficiency for the parameters such as the pump’s rotational speed (n), operating pressure (p), capacity (Q) and moment (M)?

Author Response

Dear reviewer, what is possible has been taken into account (improvement was made by the authors in the submitted article). Comments on indicated reviewer's points are sent in the attachment

Yours faithfully

Marian Partyka
Maria Natorska

Reviewer 2 Report

This is a very interesting paper on an optimization process for optimizing a gear pump. My only issue with the paper as that the authors need to explain a bit more about what they are doing. Please explain tooth root undercutting as many readers will not be familiar with this. Also, discuss various types of optimization approaches and how the current approach is an improvement over others. They need to do this in the paper and expand on it in the conclusions as well.

Author Response

Dear reviewer, what is possible has been taken into account (improvement was made by the authors in the submitted article). Comments on indicated reviewer's points are sent in the attachment Yours faithfully

Round 2

Reviewer 1 Report

none